# Latent Forcing: Reordering the Diffusion Trajectory for Pixel-Space Image Generation

Alan Baade [1]  Eric Ryan Chan [1]  Kyle Sargent [1]  Changan Chen [1]  Justin Johnson [2]  Ehsan Adeli [1]  Li Fei-Fei [1]

## Abstract

Latent diffusion models excel at generating high-quality images but lose the benefits of end-to-end modeling. They discard information during image encoding, require a separately trained decoder, and model an auxiliary distribution to the raw data. In this paper, we propose Latent Forcing, a simple modification to existing architectures that achieves the efficiency of latent diffusion while operating on raw natural images. Our approach orders the denoising trajectory by jointly processing latents and pixels with separately tuned noise schedules. This allows the latents to act as a scratchpad for intermediate computation before high-frequency pixel features are generated. We find that the order of conditioning signals is critical, and we analyze this to explain differences between REPA distillation in the tokenizer and the diffusion model, conditional versus unconditional generation, and how tokenizer reconstruction quality relates to diffusability. Applied to ImageNet, Latent Forcing achieves a new state-of-the-art for diffusion transformer-based pixel generation at our compute scale. Code and checkpoints at https://github.com/AlanBaade/LatentForcing.

## 1. Introduction

Generation best starts with high-level structure before low-level detail: Buildings are planned before construction, and movies are storyboarded before shooting. In image diffusion models, sampling conditional information such as class labels (Brock et al., 2019) or text conditions (Ramesh et al., 2021) generally precedes the generation of visual content.

[1]Department of Computer Science, Stanford University, California, USA [2]Department of Computer Science and Engineering, University of Michigan, Michigan, USA. Correspondence to: Alan Baade <baade@stanford.edu>.

*Proceedings of the 43rd International Conference on Machine Learning*, Seoul, South Korea. PMLR 306, 2026. Copyright 2026 by the author(s).

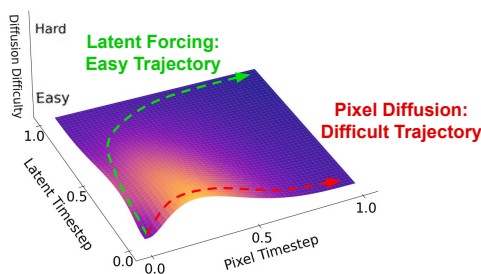

*Figure 1.* Conceptual Diagram. We diffuse both latents and pixels together, each with their own time variable. This allows us to denoise an easier trajectory than pure pixel diffusion by diffusing latents first, leading to improved performance.

Moreover, the generation order of visual content is itself a large design space: discrete generative models have explored raster (Brock et al., 2019) and random (Li et al., 2024) orders, while pixel-space diffusion models perform autoregression in the frequency domain (Dieleman, 2024).

The current state-of-the-art for visual generation leverages latent "tokenizers" (Esser et al., 2020; Rombach et al., 2021b; Peebles & Xie, 2023), which downsample visual data to a learned latent space where the final generative model is trained, and are trained in a separate, preliminary stage. Although this multi-step design improves generation quality, it sacrifices some of the benefits of end-to-end modeling, as the encoder destroys information from the data distribution, and a cascaded decoder is required at generation. As a result, a complicated dilemma arises from the interaction between tokenizers and the downstream generative model: more lossy tokenization may permit generative models to converge faster but with a lower ceiling to overall performance, while more lossless tokenizers yield slower convergence but a higher performance ceiling. Moreover, information loss in the tokenization pipeline results in poor reconstruction of human-salient features such as faces and text (Yao et al., 2025). To address these issues, discovering high-quality latent encoders and decoders that support the efficient training of diffusion models while maintaining high-quality reconstruction has been the focus of extensive research (Yang et al., 2025; Kouzelis et al., 2025a).

In contrast to latent diffusion models, pixel-space diffusion models attempt to learn diffusion on natural images directly.

Although falling out of favor in the past few years, pixel-space approaches have recently made significant improvements. In particular, JiT (Li & He, 2025) demonstrates that diffusion can operate in high-dimensional spaces by changing the loss prediction target from velocity prediction to direct prediction of the denoised target. This obviates one of the core benefits of latent diffusion: a dramatically reduced tokenizer dimension. Because of improvements in pixel space generation and their end-to-end nature, some contend that pixel space generation will eventually outscale latent approaches (Yan et al., 2025) to create a final, end-to-end simplified pipeline for generative modeling.

In this paper, we reconsider latent and pixel diffusion models in terms of the order of the information that they generate. Specifically, we reevaluate commonly understood assumptions about latent diffusion models, such as the reconstruction-generation tradeoff, (Yao et al., 2025), in terms of this ordering process. Metrics commonly used as proxies for the "diffusability" (Skorokhodov et al., 2025) of a tokenizer space, such as compression rates, are *global* properties of the tokenizer. Instead, we take a *timestep-specific* view: What if instead of overall latent space compression, only early timesteps need to be compressed, leaving later timesteps to maintain high-level detail? In fact, this ordering is already implicitly assumed in latent diffusion pipelines: The diffusion model generates coarse structure, and the decoder, whether a GAN or a diffusion model, renders the image details.

Simultaneously, in this view, it may also be the case that pixel diffusion approaches are *too* inflexible. The ground truth denoising process of natural images is governed by their frequency distribution (Dieleman, 2024). Thus, pixel-space diffusion invariably means predicting low-frequency details before high-frequency details, rather than predicting more helpful information such as semantics first.

To reconcile latent and pixel diffusion, we propose Latent Forcing. In Latent Forcing, we train a single diffusion model over a pixel space and latent space simultaneously with multiple time variables. By scheduling the denoising trajectory to reveal self-supervised encoder latents before pixels, we achieve the convergence benefits of latent diffusion without losing information due to a tokenizer. The generated latent, which effectively serves as a "scratchpad" to condition the generation of the natural image, is discarded at the end of denoising process. Our approach is built for simplicity and follows the principles that have been shown to scale: We use a standard diffusion transformer (Peebles & Xie, 2023), with matched compute to existing approaches.

Additionally, we explore ordering as an explanation for several commonly observed behaviors in diffusion models, including the source of benefits from incorporating self-supervised encoders into the diffusion versus tokenization process, and the differences between conditional and unconditional generation. We isolate that order, rather than distillation alone, is a primary factor driving the effectiveness of pretrained representations in tokenization.

In summary, we contribute the following:

- We introduce Latent Forcing, a pixel-space generation approach with the benefits of latent diffusion.
- We analyze the importance of ordering latent versus pixel data in the diffusion trajectory, finding that it is a driving factor for performance in both conditional and unconditional generation.
- We empirically validate our method and analysis on ImageNet (Deng et al., 2009), obtaining state-of-the-art results on conditional and unconditional generation for pixel diffusion transformers at our compute scale.

## 2. Related Work

**Diffusion Models** (Sohl-Dickstein et al., 2015; Ho et al., 2020) are a general, scalable technique for generative modeling. Flow-based approaches (Liu et al., 2022) simplify the diffusion training and inference pipeline and have extensive relations to diffusion models (Salimans & Ho, 2022). We use diffusion and flow interchangeably in this work.

**Latent Diffusion.** Diffusion modeling in a learned latent space is the current state-of-the-art for visual generation. The earliest approaches for latent diffusion used continuous embeddings from KL-regularized autoencoders (Kingma & Welling, 2013). VQGAN (Esser et al., 2020) and (Rombach et al., 2021a) largely defined the currently existing paradigm of latent diffusion decoders, where a combination of a reconstruction, adversarial loss (Goodfellow et al., 2014), VAE KL loss, and perceptual loss (Johnson et al., 2016) are used to jointly train an encoder and decoder. When designing latent spaces for diffusion models, a major concern is the reconstruction-generation tradeoff (Chen et al., 2024), (Yao et al., 2025), where latents that maintain more information about the input sacrifice generation quality in the diffusion model. As a result, state-of-the-art latent generation models generally operate with low ($<32$) PSNR tokenizers. Although GAN decoders have remained popular in state-of-the-art models, diffusion based decoders have also demonstrated strong performance (Sargent et al., 2025; Chen et al., 2025b).

While the decoder has remained largely similar for several years, in the past year generation and encoding of latent diffusion models has seen strong gains driven by including representations from pretrained models such as DINOv2 (Oquab et al., 2023; Darcet et al., 2023) in the diffusion or tokenization process (Yao et al., 2025; Yu et al., 2025; Zheng et al., 2025). Using these representations, prior work (Leng et al., 2025) has attempted to make generation end-

to-end by aligning with these pretrained models, however these approaches still fall short of being end-to-end, relying on lossy encoders and separately trained GAN decoders. An alternate line of work has looked at direct regularization of the latent space to improve *diffusability* (Kouzelis et al., 2025a; Skorokhodov et al., 2025; Yang et al., 2025). ReDi (Kouzelis et al., 2025b) is closely related to our work and explores jointly training on DINOv2 and VAE latent spaces. In our work we study the diffusability of pixel-space generation by separately diffusing latent conditions and then diffusing pixels with these conditions.

**Pixel Diffusion Models.** The earliest applications of diffusion models to image generation denoised in the pixel space. This simplifies the pretraining objective, but often came at the expense of architectural complexity, such as U-Net architectures (Ronneberger et al., 2015). Other works such as Matryoshka Diffusion (Gu et al., 2024) and Hourglass Transformers (Crowson et al., 2024) explored pyramidal or hierarchical representations to ease learning in pixel space. Simple diffusion (Hoogeboom et al., 2023) dramatically improved pixel space generation by investigating the importance of the noise schedule. Recently, JiT (Li & He, 2025) demonstrated that by modifying the training output to predict denoised pixels directly, simple transformer architectures can work at predicting high-dimension data, provided the data sits on a low-dimensional manifold like images. In our work, we build on JiT to create a new state-of-the-art transformer pixel diffusion approach.

**Time Scheduling in Diffusion.** The time schedule is a critical choice in efficiently training diffusion models (Karras et al., 2022), needs to be adjusted alongside image resolution (Hoogeboom et al., 2023), and should create outputs with appropriate loss weighting (Hang et al., 2023). Closely related to our work, Diffusion Forcing (Chen et al., 2025a) demonstrates that incorporating multiple time schedules into training allows one to modify the diffusion process while still optimizing for the ELBO of the input data. However, Diffusion Forcing is formulated for autoregressive modeling of temporally-ordered data, and still requires tokenization at both encoding and decoding. In contrast, we investigate multi-time diffusion in a non-autoregressive setting, jointly generating pixels and deterministic latent representations of the same input.

**Generation Order.** The ordering of generation signal has been shown to be very important in discrete diffusion models both for images (Besnier et al., 2025) and text token order in masked generation models (He et al., 2022), often in the context of increasing the independence of concurrently generated samples. One especially important theoretical property of modifying information order in diffusion generative models is that different conditioning orderings can lead to exponential differences in the learnability of data

(Gupta et al., 2024).

Generating latent structure as conditioning has also demonstrated improvements in both diffusion and text. Representation-Conditioned Generation (RCG) (Li et al., 2023) showed that generating the CLS token of pretrained image models for guidance conditioning can improve both conditional and unconditional generation. In our paper, we take representation-conditioning even further, generating full latent representations and removing extra architectural components. Joint latent generation with text has also improved reasoning in autoregressive language models, where latents can act as scratchpad chain-of-thought reasoning to output an answer (Hao et al., 2024).

# 3. Ordering the Diffusion Process

Image diffusion models traditionally use one global time variable and one token space. The core idea of our paper is to diffuse multiple modalities (e.g. DINOv2 (Oquab et al., 2023) and Pixels), each with their own time variable. Then, at both training and inference, we schedule each modality's time variable to denoise in a manner that leads to optimal performance. In this section, we describe how to order the diffusion process, and in the next section we instantiate Latent Forcing models on ImageNet (Deng et al., 2009).

## 3.1. Flow-Based Diffusion Review

We establish notation by briefly reviewing Flow-Based Diffusion Models (Liu et al., 2022) with one modality and time variable. For our input data distribution $\mathbf{x} \sim p_{\text{data}}$ and noise distribution $\boldsymbol{\epsilon} \sim p_{\text{noise}} = \mathcal{N}(0, \mathbf{I})$, the noised latent at timestep $t \in [0, 1]$ is denoted $z_t = t\mathbf{x} + (1 - t)\boldsymbol{\epsilon}$. We use $t = 1$ when $z_t$ is pure data and $t = 0$ when $z_t$ is noise. The diffusion model, $\mathbf{v}_\theta$, trains to minimize $\mathbb{E}_{\mathbf{x}, \boldsymbol{\epsilon}} \|\mathbf{v}_\theta(z_t, t) - (\mathbf{x} - \boldsymbol{\epsilon})\|^2$ for a given $t$. In practice, we follow JiT (Li & He, 2025) and implement $\mathbf{v}_\theta$ as a $\mathbf{v}$-loss with $\mathbf{x}$ prediction to avoid the information capacity constraint of predicting $\boldsymbol{\epsilon}$ or $\mathbf{v}$ directly.

## 3.2. Flow on Multiple Tokenizers

We can similarly define diffusion on $k$ modalities and time variables, where in practice we use $k = 2$. For a sequence of inputs $\{\mathbf{x}_i \sim p_{\text{data}_i}\}_{i=1}^k$ and times $\{t_i \in [0, 1]\}_{i=1}^k$, we noise $\{\mathbf{x}_i\}_{i=1}^k$ with $\{\boldsymbol{\epsilon}_i\}_{i=1}^k$, giving $\{\mathbf{z}_{i,t_i}\}_{i=1}^k$. Let $\mathbf{v}_\theta(\cdot) = [\mathbf{v}_{\theta,1}, \dots, \mathbf{v}_{\theta,k}]$ be the outputs of our model corresponding to $\{\mathbf{x}_i\}_{i=1}^k$. We train to minimize

$$\mathcal{L} = \sum_{i=1}^k \lambda_i \mathbb{E} \|\mathbf{v}_{\theta,i}(\mathbf{z}_{1,t_1}, \dots, \mathbf{z}_{k,t_k}, t_1, \dots, t_k) - (\mathbf{x}_i - \boldsymbol{\epsilon}_i)\|^2 \tag{1}$$

where expectations are over all random variables, $t_i$ may be correlated, and $\lambda_i$ are loss weights.

To denoise at inference, we set a global time value $t_{\text{global}}$ and define the per-modality schedules as $t_i = f_i(t_{\text{global}})$ on $[0, 1]$. We require $f_i$ to be non-decreasing with $f_i(0) = 0$ and $f_i(1) = 1$. Then, for an Euler step from global time $t$ to $s$, we perform

$$\mathbf{z}_{i,f_i(s)} = \mathbf{z}_{i,f_i(t)} + (f_i(s) - f_i(t)) \cdot \mathbf{v}_{\theta,i}(\cdot) \quad (2)$$

The mutual information of a latent with respect to the noised latent $I(\mathbf{x}_i; \mathbf{z}_{i,t_i})$ is strictly monotonic with respect to noise and is upper-bounded by the Gaussian channel capacity $\frac{1}{2}\log_2(1 + \text{SNR}_i)$, where SNR is the Signal-to-Noise Ratio. Therefore, we define the generation order as the trajectory of SNRs for each noised latent during generation, a proxy for the relative rate at which information is revealed across different tokenizers:

$$\mathcal{O}(t_{\text{global}}) = \left( \frac{f_i(t_{\text{global}})^2 \mathbb{V}[\mathbf{x}_i]}{(1 - f_i(t_{\text{global}}))^2} \right)^k_{i=1} \quad (3)$$

As shown in prior work (Chen et al., 2025a), denoising multiple modalities $X, Y$ at different time schedules is a valid model for the joint distribution $P(X, Y)$. In this paper, we focus on generating latents $Y$ output by a deterministic function of the pixels of $X$, such as DINOv2 features. With non-overlapping time schedules, the generation process factors as $P(Y)P(X|Y)$. When modeling a deterministic latent, $P(Y|X) = 1$, this process optimizes for the probability of the raw data regardless of ordering $P(X, Y) = P(Y|X)P(X) = P(X)$. As a result, our approach differs from recent diffusion tokenization work such as ReDi (Kouzelis et al., 2025b), which attributes benefits of latent tokenization to modeling a joint distribution. Instead, our work suggests that latents help generation without modeling a joint distribution at all, where $P(\text{Pixels}, \text{Latents})$ reduces purely to $P(\text{Pixels})$.

### 3.3. Scaling as Time Scheduling

Because SNR is a ratio between the variance of the input and the variance of the noise, scaling a data point $x_i$ is informationally equivalent to shifting the noise schedule, and this relationship between data magnitudes and noise scale is well understood (Hoogeboom et al., 2023). As a corollary, when combining multiple modalities, scaling the variance of each modality implies changing the order of generation. Therefore, combining multiple tokenizers implicitly involves choosing a time schedule per tokenizer, regardless of whether we use additional time variables.

As discussed in prior work (Esser et al., 2024), the time shift function that is informationally equivalent to scaling the magnitude of latent $\mathbf{x}$ by scalar $\alpha$ is:

$$f_{\alpha\text{-shift}}(t) = \frac{t\alpha}{1 + (\alpha - 1)t} \quad (4)$$

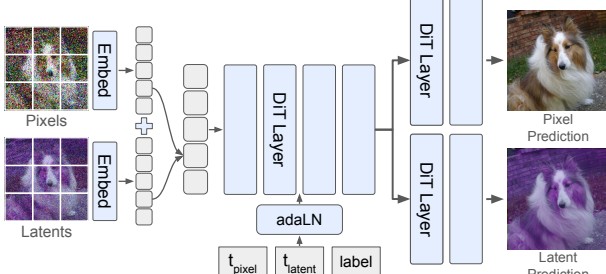

*Figure 2.* Model Architecture. Our approach makes minimal changes to diffusion transformers. Left: We add the per-patch embeddings of latents and pixels together, keeping the same number of tokens. Middle: We use two time variables instead of one for adaLN. Right: Optionally, we take the last $M = 4$ transformer layers and split them into two $M/2$-layer output experts.

$f_{\alpha\text{-shift}}$ is derived by holding the SNR constant under scaling, making it a natural function to explore for time scheduling.

## 4. Latent Forcing

We now explore Latent Forcing empirically for the task of diffusion transformer (DiT) (Peebles & Xie, 2023) pixel generation on ImageNet (Deng et al., 2009).

### 4.1. Tokenization

For all experiments, we focus on joint generation of two modalities: pixels and latent embeddings. By default, we follow prior work (Yao et al., 2025; Yu et al., 2025) and use DINOv2 (Oquab et al., 2023) for the latent space. We also experiment with Data2Vec2-Large (D2V2) (Baevski et al., 2023), a self-supervised model trained for only 150 epochs on ImageNet, and spatially downsampled $64 \times 64$ pixel images, commonly used to bootstrap high-resolution pixel generation (Ho et al., 2022).

For an input image $I \in \mathbb{R}^{256 \times 256 \times 3}$, we first obtain pixel representations by patchifying into 256 tokens, using a patch size of 16: $x_{\text{pixel}} \in \mathbb{R}^{16 \times 16 \times 768}$, where $768 = 16 \cdot 16 \cdot 3$ for three color channels and patch size 16. Like REPA (Yu et al., 2025), we construct our latent space such that latent patches align with pixel patches. For DINOv2, which uses a $14 \times 14$ patch size, we resize the image to $224 \times 224$ before encoding. For Data2Vec2, which trains on $224 \times 224$ images with a patch size of $16 \times 16$, we interpolate to upsample position embeddings. At the output of our latent embedding model, we then have $x_{\text{latent}} \in \mathbb{R}^{16 \times 16 \times D}$, where $D$ is the latent dimension. We normalize pixels to $[-1, 1]$, and we rescale the latent embeddings to match the global variance of the normalized pixels. To decode into an image, we discard the generated latents and renormalize the output pixels.

### 4.2. Architecture

Latent Forcing requires minimal architecture changes to standard diffusion transformers. In traditional DiTs on Ima-

geNet, adaLN-Zero (Peebles & Xie, 2023) adds a learned class embedding to a time embedding for conditioning, where the time embedding is output from a two-layer MLP. To condition on two time variables, we add a second time embedding MLP, which increases the parameter count by roughly 0.5%. For tokenizer input, we follow JiT (Li & He, 2025) and use a 128-dimensional linear bottleneck to project each latent into an embedding equal to the transformer hidden size. To combine multiple tokenizers, we simply add these embeddings together, leading to the **same number of tokens** and nearly-identical training and inference speed. For tokenizer output, we use a linear projection per latent.

We also explore one optional change to improve performance, where we take the last 4 transformer layers of our model and split them into 2 output experts, one for latent output and one for pixel output. We do this because the default approach of only applying linear projections from the final embeddings of our model to multiple outputs may strain network capacity. This modification adds no parameters and uses identical FLOPs. Our architecture is shown in Figure 2. Unless otherwise stated, all other architectural decisions match JiT, which, like LightningDiT (Yao et al., 2025), takes advantage of the generalizability of the transformer architecture to build on advancements from the wider research community such as RoPE (Su et al., 2024). By default, we train with a size of ViT/L for 80 Epochs.

### 4.3. Prediction Target

As analyzed extensively in JiT (Li & He, 2025), predicting in high dimensional output spaces performs catastrophically when predicting noisy output targets. Therefore, for all experiments we follow JiT and use $\mathbf{x}$-prediction with $\mathbf{v}$-loss weighting, where $t_{\text{clip}} = 0.05$ prevents dividing by zero:

$$\mathcal{L} = \left\| \frac{(\mathbf{x}_{\text{pred}} - \mathbf{z}_t)}{\max(1 - t, t_{\text{clip}})} - \frac{(\mathbf{x} - \mathbf{z}_t)}{\max(1 - t, t_{\text{clip}})} \right\|^2 \quad (5)$$

### 4.4. Training Time vs Inference Time Ordering

We investigate two model types for latent forcing. To explore the space of possible orderings, we first implement a model that trains on independently sampled time variables per modality. With this, we can sample in any order at inference time, and we call this model the "Multi-Schedule Model." Second, in what we refer to as the "Single-Schedule Model," we fix a global time variable $t_{\text{global}}$, and define latent specific time variables as a function of $t_{\text{global}}$, following Sec 3.2. This means that each Single-Schedule Model uses a single fixed training and inference trajectory. Although the Multi-Schedule Model allows for any inference trajectory that a Single-Schedule Model may follow, at inference time we commit to a single inference trajectory. Therefore, given a compute and parameter budget, we would expect a Single-Schedule Model that trains exclusively in-distribution to

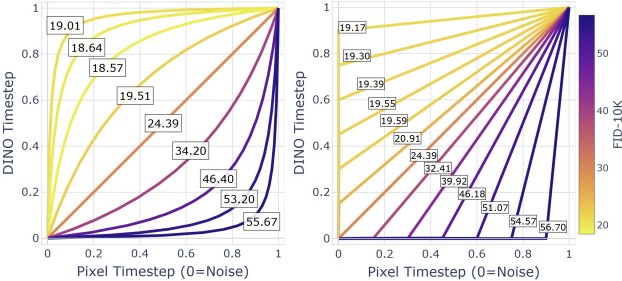

*Figure 3.* FID-10K values for different diffusion trajectories through a joint DINOv2 and Pixel Space.

the inference schedule to perform better. Because of this, we use the Multi-Schedule Model to explore the space of ordering while holding model quality constant, while for baseline comparisons we use the Single-Schedule Model.

### 4.5. Multi-Schedule Diffusion

To implement the Multi-Schedule Model, we first establish a training time schedule. Traditional diffusion transformers follow a shifted logit-normal schedule (Karras et al., 2022). However, when sampling multiple variables this would result in a product distribution where certain inference trajectories, such as a cascaded schedule that entirely denoises latents before pixels, would receive zero training signal. To resolve this, we instead sample from the uniform distribution and apply a time shift (Equation 4). As discussed in Sec 3.3, a time shift is equal to scaling the latent input, so a uniform distribution when the tokenizer has variance $\alpha$ is informationally equal to shifted sampling when the variance is 1. Therefore, we apply a time shift for pixels and latent features such that timestep sampling is uniform when both the pixel and latents have variance 1. To balance the gradient magnitude at low-noise timesteps, we set $t_{\text{clip}} = 1/3$ (Eq. 5). Finally, for all models we set the loss weights ($\lambda_i$ in Equation 1) such that training has equal loss magnitude for the pixel and latent space.

We demonstrate quantitative results for the Multi-Schedule Model for different trajectories in Fig 3, where we see a clear trend that latent features should denoise earlier than pixels. This result holds across multiple different curve trajectories. We also observe that this ordering is most important at early timesteps, where in Fig 3, right, a linear schedule after DINOv2 has denoised to $t_{\text{DINO}} = 0.15$ captures a majority of the gains on FID (Heusel et al., 2017) from ordering.

In Table 1, we explore this ordering for additional latent spaces. Similarly to cascaded diffusion approaches (Ho et al., 2022), we find that generating downsampled pixels before denoising the full-resolution image leads to benefits in generation. Furthermore, we see that the benefits from ordering latent structure holds across different latent embedding models, and isn't isolated to DINOv2.

*Table 1.* FID scores for Multi-Schedule Models across different schedules. Schedules follow $t_{\text{latent}} = f_\alpha(t_{\text{global}}), t_{\text{pixel}} = t_{\text{global}}$ from Eq. 4

| | PIXEL EARLIER | | | EQ. SNR | LATENT EARLIER | | |
|---|---|---|---|---|---|---|---|
| LATENT MODEL | 1/64 | 1/16 | 1/4 | 1 | 4 | 16 | 64 |
| $64 \times 64$ PIXELS | 44.51 | 44.45 | 44.35 | 44.57 | 44.20 | 42.35 | **42.31** |
| DATA2VEC2 | 55.19 | 50.24 | 38.24 | 27.69 | 24.26 | **23.61** | 24.44 |
| DINOV2-B+REG | 55.35 | 50.64 | 37.63 | 24.39 | 18.99 | **18.65** | 18.90 |

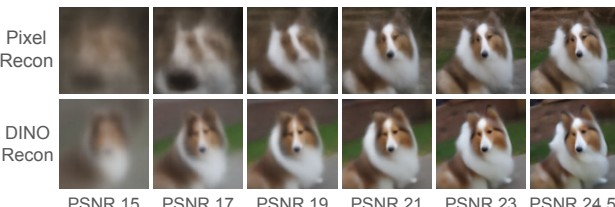

Pixel Recon

DINO Recon

PSNR 15   PSNR 17   PSNR 19   PSNR 21   PSNR 23   PSNR 24.5

*Figure 4.* Output single-step **x**-predictions in the pixel space from the Multi-Schedule Model, where each column reconstructs with the same PSNR. Top: Predictions when pixel features are partially denoised, and DINOv2 features are fully noised. Bottom: Predictions when latent DINOv2 features are partially denoised and pixels are fully noised. At low PSNR levels, DINOv2 features preserve significantly more spatial information.

Qualitative results for the Multi-Schedule model can be seen in Figure 4, where we compare single step class conditioned generation results at different DINOv2 and pixel timesteps, organized by the PSNR on pixels. Even though both reconstructions have the same MSE on the ground truth image, the DINOv2 predictions maintain significantly more structural features. Meanwhile, the pixel generation component demonstrates large-scale structural uncertainty. We hypothesize that this structural difference in denoised predictions is closely related to why some orders are better than others.

Finally, we inspect how reconstruction interacts per timestep in Figure 5, where we plot the PSNR on Pixels and DINOv2 features across timesteps in $\mathbb{R}^{[0,1] \times [0,1]}$. We observe that pixel features more sharply increase in PSNR at early timesteps when no DINOv2 information is present. We also see that pixel features improve predictions on DINOv2 at nearly all noise levels, whereas DINOv2 features do not inform pixel generation when $t_{\text{pixel}} \geq 0.75$.

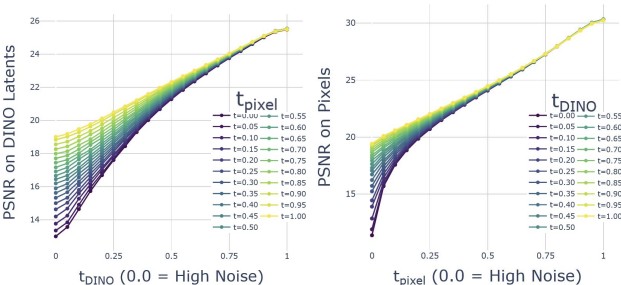

*Figure 5.* PSNRs on DINOv2 and Pixel features at different timestep combinations in the Multi-Schedule Model.

*Table 2.* Ablation for the probability of sampling a latent timestep. FID-10K without guidance.

| $P_{\text{LATENT}}$ | 0.3 | 0.4 | 0.5 | 0.6 | 0.7 | 0.8 |
|---|---|---|---|---|---|---|
| FID ↓ | 12.54 | **12.42** | 12.77 | 13.17 | 13.48 | 16.13 |

*Table 3.* Ablation: Logit schedule mean for latent timesteps, using DINOv2. FID-10K.

| $\mu_{\text{LATENT}}$ | FID UNGUIDED↓ |
|---|---|
| 1.0 | 14.11 |
| 1.2 | **13.97** |
| 1.4 | 14.41 |

*Table 4.* Ablation: Logit schedule mean for pixel timesteps. FID-10K.

| $\mu_{\text{PIXEL}}$ | FID UNGUIDED↓ |
|---|---|
| 0.0 | 18.28 |
| 0.4 | **16.13** |
| 0.8 | 16.38 |

## 5. Single-Schedule Modeling

As observed in our Multi-Schedule Model experiments, generating latent features before pixel features strongly outperforms predicting pixel features first. However, it's possible that these results may not carry over to a model trained for a single trajectory. In this section, we commit to a single ordering for both training and inference time.

### 5.1. Time Sampling

Time schedule weighting is essential for diffusion transformers (Karras et al., 2022), and adding a second time parameter dramatically increases the potential search space for an optimal weighting. To simplify this search space, we first focus on training on a cascaded schedule where the latent generates entirely first, and pixels generate afterward.

For cascaded generation, we split time selection into first choosing whether to optimize for latent denoising or pixel denoising with probability $p_{\text{latent}}$, and after that we sample the noise level. Following (Zheng et al., 2025) for latent features and (Li & He, 2025) for pixel features we use a logit-normal schedule for each input space, with separately tuned parameters. We show ablations for $p_{\text{latent}}$ in Table 2 and for logit-normal parameters in Tables 3 and 4. When sampling a latent step, we set $t_{\text{pixel}} = 0$ (full noise) and disable the loss on pixels. When sampling pixels, we by default set $t_{\text{latent}} = 1.0$ (no noise) and disable the loss on the latent. To minimize gradient variance, we choose loss weights $\lambda$ such that latent and pixel losses are equal, instead implicitly weighting them with $p_{\text{latent}}$.

### 5.2. Improving Cascaded Generation

Traditional image diffusion models are weakly conditioned at high noise levels, learning solely to output a global or class-conditioned image average independent of the noised input. However, in Latent Forcing, denoised latent structure may strongly condition the pixel generation at $t = 0$, making the function the model needs to learn difficult. This means that a logit-normal schedule, which has zero probabil-

*Table 5.* Ablations for the two modeling choices in Latent Forcing that deviate from the standard DiT pipeline. FID-10K Unguided.

| MODEL | FID UNGUIDED ↓ |
|---|---|
| LF-DiT-L | **12.42** |
| −OUTPUT EXPERT LAYERS | 12.93 |
| −10% EARLY TIMESTEPS | 12.98 |

*Table 6.* Max noise applied to latent features during pixel timesteps vs FID-10K w/o guidance. Top: Without noise FID increases when training longer. Bottom: Adding small noise fixes this issue.

| NOISE-$\beta$ | EPOCHS | FID↓ |
|---|---|---|
| 0% | 80 | 13.97 |
| 0% | 200 | 16.47 |
| 25% | 80 | 13.48 |
| 25% | 200 | **10.93** |
| 50% | 80 | 14.64 |

*Table 7.* Inference noise at 200EP, when training with max 25% noise ($t > 0.75$). Despite noise improving cascaded generation at training, it's harmful at inference.

| NOISE | FID |
|---|---|
| 0% | **10.93** |
| 5% | 11.07 |
| 10% | 11.46 |
| 15% | 11.81 |
| 25% | 12.55 |

*Table 8.* FID-10K scores for different Single-Schedule Model time schedules. A cascaded schedule performs best, and joint denoising according to a variance shift (Equation. 4) works well.

| TIME SCHEDULE | FID UNGUIDED↓ | FID GUIDED |
|---|---|---|
| CASCADED | **12.42** | **6.60** |
| LIN. OFFSET $o = 0.1$ | 20.98 | 10.73 |
| VAR. SHIFT $\alpha = 9$ | 13.48 | 8.16 |

ity mass at $t_{\text{pixel}} = 0$, may harm performance. We ablate this concern in Table 5, where 10% of the time we sample the pixel timestep in $U[0, 0.5]$ instead of from the logit-normal distribution, and find that this decision moderately improves performance.

Cascaded generation is also known for cascaded error, where errors in earlier generation compound into later outputs. We observe a version of this error in Latent Forcing, which we show in Table 6. To address this, we modify the cascaded schedule such that the latents receive minor noise during pixel steps, $t_{\text{latent}} \in U[1 - \beta, 1]$, where $\beta$ is the maximum amount of noise added. Interestingly, we find that noise is only helpful at training, seen in Table 7. Combined with the performance degrading later into training, we interpret this improvement as an augmentation that prevents the model from overfitting to high-frequency, difficult-to-generate details in the latent space when generating pixels.

### 5.3. Time Schedules

We now expand to training latent forcing at more general time schedules. First, we implement a Variance Shifted schedule where $t_{\text{latent}} = f_\alpha(t_{\text{global}})$, informationally equivalent to linearly scaling up the latent features by $\alpha$, and $t_{\text{pixel}} = t_{\text{global}}$. This variance schedule performs best according to our Multi-Schedule Model experiments, and we use the optimal value $\alpha = 9$, with an FID-10K unguided of 18.57. Second, we implement a Linear Offset schedule where both time variables advance linearly, however $t_{\text{pixel}}$ is delayed to start at offset $o$. Examples of these schedules are visualized in Figure 3.

We follow the cascaded model and sample the time schedule per latent space. When sampling timestep $t_i$, we obtain the corresponding timestep $t_j$ according to the given schedule. We do this by defining $g_i$ to be $f_i$ domain restricted to where the schedule is advancing during inference, $f'_{(.)}(t_{\text{global}}) > 0$, meaning $g$ is strictly monotonic and invertible. Then, we convert from the sampled timestep $t_i$ to a different latent timestep $t_j$ using $t_j = g_j(g_i^{-1}(t_i))$, where $f_\alpha^{-1} = f_{1/\alpha}$.

Results for different time schedules are presented in Table 8. For all generation techniques, we use 50 Heun steps, where at inference time we equally space timesteps along the $(f_{\text{pixel}}(t_{\text{global}}), f_{\text{latent}}(t_{\text{global}}))$ trajectory. For cascaded generation, this results in 25 latent timesteps followed by 25 Pixel timesteps. We find that cascaded generation has the best performance, however jointly denoising multiple tokenizers at once according to a variance shift is also performant.

### 5.4. Guidance

For all models, we compare both with and without guidance. We implement both AutoGuidance (Karras et al., 2024) and Classifier Free Guidance (Ho & Salimans, 2022) for every model we train and evaluate. Similar to RAE, we find that AutoGuidance performs best for Latent Forcing, which we attribute to DINOv2 features probing to the class label, making class conditioning redundant during pixel generation timesteps. We report CFG restricted to an interval (CFG-Interval) (Kynkäänniemi et al., 2024) results in our system-level comparison (Tab. 11), where we find applying CFG-Interval for DINOv2 timesteps and AutoGuidance for pixel timesteps is best. We further discuss guidance implementation details in Appendix A.2

### 5.5. Distillation vs Ordering

REPA (Yu et al., 2025) demonstrated remarkable gains in diffusion model performance and started a large line of research into how to best incorporate externally pretrained representations into diffusion models. However, REPA has been shown to lose effectiveness during late-stage training (Wang et al., 2025), and the necessity of using externally pretrained representations has been challenged by methods that incorporate additional losses into diffusion modeling (Wang & He, 2025). An alternate view on REPA, then, is that the gains from REPA-distillation are merely one-time benefits that speed up training but ultimately distract from the underlying objective of generative modeling that has

*Table 9.* FID-50K scores for conditional generation at 80 epochs.

| MODEL | FID UNGUIDED↓ | FID GUIDED↓ |
|---|---|---|
| JIT | 25.18 | 5.64 |
| JIT+REPA | 18.60 | 4.57 |
| LF-DIT DINOV2 | **9.76** | **4.18** |
| LF-DIT D2V2 | 12.46 | 5.45 |

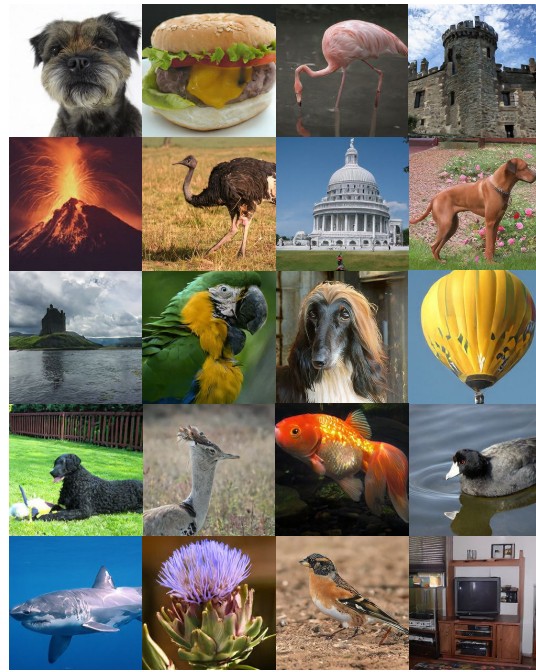

*Figure 6.* Curated Results from LF-DiT-L at 200 epochs, using AutoGuidance with $\omega = 1.5$.

been shown to scale.

In Table 9, we compare Latent Forcing (LF-DiT) against the state-of-the-art pixel diffusion transformer, JiT, as well as JiT enhanced with REPA. We demonstrate that the gains from ordering are distinct from distillation, with a $1.9\times$ reduction in unguided FID-50K versus JiT with REPA, and a $2.5\times$ reduction compared to JiT.

### 5.6. Conditioning as Ordering

Our paper focuses on a reordering of the bits of information during the generative process, and has found that treating ordering as a first-class citizen during generation can lead to stronger performance. However, for all previous experiments, the first information about the target distribution during the generation process was not from the diffusion process but was instead from the class conditioning. In this section, we explore unconditional generation, where ordering is determined *purely* by the tokenizer. To implement unconditional generation, we apply no changes to our model and sample only the "no-class" label during training.

We report results for unconditional generation in Table 10,

*Table 10.* FID-50K for unconditional generation at 80 epochs.

| MODEL | FID UNGUIDED↓ | FID GUIDED↓ |
|---|---|---|
| JIT | 53.26 | 44.80 |
| JIT+REPA | 35.04 | 24.40 |
| LF-DIT DINOV2 | **20.44** | **13.36** |
| LF-DIT D2V2 | 20.99 | 15.56 |

*Table 11.* System-level comparison of pixel and latent diffusion models, with latent diffusion approaches sorted by PSNR to show the information lost during encoding. FID-50K Results on ImageNet 256×256, (U) Unguided, (G) Guided. Citations: RAE (Zheng et al., 2025), DiT (Peebles & Xie, 2023), SD-VAE (Rombach et al., 2021b), SiT (Ma et al., 2024), REPA (Yu et al., 2025), MAR (Li et al., 2024), Detok (Yang et al., 2025), LightningDiT (Yao et al., 2025), REPA-E (Leng et al., 2025), Uniflow (Yue et al., 2025), ADM (Dhariwal & Nichol, 2021), SiD (Hoogeboom et al., 2023), SiD2 (Hoogeboom et al., 2024), JiT (Li & He, 2025).

| MODEL | PARAMS | DEC. PARAMS | EPOCHS | PSNR↑ | FID(U)↓ | FID(G)↓ |
|---|---|---|---|---|---|---|
| *Latent Diffusion* | | | | | | |
| RAE | 839M | 415M | 800 | 18.09 | 1.51 | 1.13 |
| DIT-XL/2+SD-VAE | 675M | 49M | 1400 | 23.40 | 9.62 | 3.04 |
| SIT-XL/2+SD-VAE | 675M | 49M | 1400 | 23.40 | 8.3 | 2.62 |
| REPA+SD-VAE | 675M | 49M | 800 | 23.40 | 5.9 | 1.42 |
| MAR + DETOK | 479M | 86M | 800 | 24.06 | 1.86 | 1.35 |
| LIGHTNINGDIT | 675M | 41M | 800 | 25.29 | 2.17 | 1.35 |
| REPA-E | 675M | 41M | 1480 | 26.25 | 1.69 | 1.12 |
| MAR + UNIFLOW | 479M | 300M | 400 | 32.48 | 2.45 | 1.85 |
| *Pixel Diffusion* | | | | | | |
| ADM | 554M | 0 | 400 | ∞ | 10.94 | 3.94 |
| SID UVIT/2 | 2B | 0 | - | ∞ | 2.77 | 2.44 |
| SID2 UVIT/2 | - | 0 | - | ∞ | - | 1.73 |
| *ViT Pixel Diffusion* | | | | | | |
| JIT-L | 459M | 0 | 200 | ∞ | 16.21 | 2.79 |
| LF-DIT-L | 465M | 0 | 200 | ∞ | 7.2 | 2.48 |

where we see that ordering the latent space is critical for performance. In unconditional generation, the improvement between our approach and REPA-distillation can be further seen with a $1.8\times$ decrease in guided FID, showing that ordering, not just externally pretrained features or additional losses, is critical for generation. We further observe that these improvements are possible across multiple tokenizers, where unconditional generation using Data2Vec2 (Baevski et al., 2023) outperforms distillation from DINOv2, even though Data2Vec2 trains for only 150 epochs on ImageNet.

### 5.7. Discussion: Towards Rethinking Compression

To the best of our knowledge, Latent Forcing is the *least-ever* compressed input space for ImageNet-256, using six floats per pixel in our default configuration with DINOv2. Despite this, we outperform existing pixel-space approaches in both conditional and unconditional generation. In Table 11, we perform a system-level comparison against several existing pixel and latent diffusion models, grouped by reconstruction quality as measured by PSNR. Losing more information about the pixel space tends to improve generation quality. Our work instead pushes in the other direction, showing that losing information is not a requirement for improving generation, maintaining lossless reconstruction while improving diffusability.

# 6. Conclusion

It's easier to generate in some orders than others. We find that generation order is a critical and underexplored component of diffusion models, and we demonstrate that ordering the diffusion trajectory by using multiple tokenizers and time variables leads to significantly improved performance. Our approach is lossless, directly optimizes the likelihood of the pixel distribution, is end-to-end at inference, and requires minimal architectural changes to existing large scale diffusion training pipelines, making it a practical and scalable alternative to latent diffusion. We conduct extensive ablations and experiments both for any-order generation and order-specific generation, revealing that incorporating features into the tokenizer is fundamentally different from REPA-style distillation. We hope our work acts as a starting point toward rethinking the purpose of tokenization and representations for generative modeling.

## Impact Statement

This paper presents work whose goal is to advance the field of Machine Learning and Computer Vision. There are many potential societal consequences of our work, such as the potentially harmful dual use behavior of image generation.

## Acknowledgments

This material is based upon work supported by the National Science Foundation Graduate Research Fellowship under Grant No. DGE-2146755. We would also like to thank Yue Zhao for helpful discussion on the paper.

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

# A. Appendix

## A.1. Implementation Details

*Table 12.* Configuration of Latent Forcing.

|  | VIT-S | VIT-B | VIT-L |
|---|---|---|---|
| **ARCHITECTURE** | | | |
| DEPTH | 12 | 12 | 24 |
| HIDDEN DIM | 384 | 768 | 1024 |
| HEADS | 6 | 12 | 16 |
| PATCH SIZE | | 16 | |
| PIXEL BOTTLENECK | | 128 | |
| LATENT BOTTLENECK | | 128 | |
| DROPOUT | | 0 | |
| IN-CONTEXT CLS (LI & HE, 2025) | | 32 | |
| IN-CONTEXT START BLOCK | | 4 | |
| **TRAINING** | | | |
| EPOCHS | | 80 (ABLATION), 200 | |
| WARMUP EPOCHS | | 5 | |
| OPTIMIZER | | ADAM (KINGMA & BA, 2014) | |
| ADAM BETAS | | $\beta_1, \beta_2 = 0.9, 0.95$ | |
| BATCH SIZE | | 1024 | |
| LEARNING RATE | | 2E-4 | |
| LEARNING RATE SCHEDULE | | CONSTANT | |
| WEIGHT DECAY | | 0 | |
| EMA DECAY | | 0.9999 | |
| LATENT SAMPLER | | $\mu = -1.2, \sigma = 1.0$ | |
| PIXEL SAMPLER | | $\mu = -0.8, \sigma = 0.8$ | |
| CLASS TOKEN DROP | | 0.1 | |
| **SAMPLING** | | | |
| ODE SOLVER | | HEUN | |
| ODE STEPS TOTAL | | 50 | |
| ODE STEPS PIXEL (CASCADED) | | 25 | |
| ODE STEPS LATENT (CASCADED) | | 25 | |
| GLOBAL TIME STEPS | | LINEAR IN [0.0, 1.0] | |
| **PIXELS** | | | |
| NORMALIZATION | | $[-1, 1]$ LINEAR MIN-MAX RESCALING | |
| STANDARD DEVIATION (FOR RESCALING SEC 3.2) | | 0.485 | |
| LOSS WEIGHT $\lambda_{\text{PIXEL}}$ (EQ. 1) | | 1.0 | |
| **DINOV2 LATENTS** | | | |
| NORMALIZATION | | GLOBAL NORM TO $\mu = 0, \sigma^2 = 1$ PER EACH $16 \times 16 \times 768$ FLOAT, FOLLOWING RAE (ZHENG ET AL., 2025) | |
| MODEL | | DINOV2-BASE + REGISTERS (OQUAB ET AL., 2023; DARCET ET AL., 2023) | |
| LOSS WEIGHT $\lambda_{\text{LATENT}}$ (EQ. 1) | | 0.333 | |
| LAYER | | 12, PRE NORM, FOLLOWING RAE (ZHENG ET AL., 2025) | |
| **DATA2VEC2 LATENTS** | | | |
| NORMALIZATION | | GLOBAL NORM TO $\mu = 0, \sigma^2 = 1$ PER EACH CHANNEL | |
| MODEL | | DATA2VEC2-LARGE, 150EP (BAEVSKI ET AL., 2023) | |
| LOSS WEIGHT $\lambda_{\text{LATENT}}$ (EQ. 1) | | 0.25 | |
| LAYER | | 12 | |

## A.2. AutoGuidance

For AutoGuidance (Karras et al., 2024), the size of the model is critical for generation. We sweep autoguidance by training four models, ViT/S, ViT/B, ViT/S with 2 additional ViT/L Output Experts (Sec. 4.2), and ViT/B With ViT/L Output Experts. All models have a lower parameter count and FLOPS than ViT/L, leading to lower FLOPS at inference than CFG. We keep every 10 Checkpoints, sweep FID-2K to find the optimal checkpoint, and sweep FID-8K as done in JiT to find the optimal guidance schedule. Empirically, ViT/S with 2 ViT/L layers performs best, and we use the checkpoint from Epoch 40 with EMA 0.9995 for the DINOv2 Cascaded Model. For fairness, we implement AutoGuidance for JiT and JiT+REPA, however find it does not improve results compared to CFG.

## A.3. CFG on a Limited Interval

We apply CFG on a limited interval (CFG-Interval) (Kynkäänniemi et al., 2024) only for our system-level comparison against other work (Table 11). We do this to reduce the search space for CFG-Inteval with multiple tokenizers, as CFG-Interval requires extensive sweeping for latent diffusion on self-supervised encoders (Zheng et al., 2025). Specifically, we find that CFG-Interval is heavily dependent on shifting DINOv2 latents during generation (Eq 4), and we attribute this difficulty to the probing accuracy of self-supervised latents to the class embedding reducing the reliance on the class label at later diffusion timesteps compared to pixel space generation.

We find that using CFG-Interval only on latent timesteps is best, while using AutoGuidance without an interval for pixel timesteps is best. When not using interval guidance, AutoGuidance performs best for both latent and pixel timesteps, and this is the setting we report in all tables other than Table 11. For the system level comparison, we use an interval of $[0.06, 1.0]$, a guidance value of 3.0, and we perform a time shift on the inference time schedule of $\alpha = 0.575$ for DINOv2. For pixel features (and for latent features in all other experiments), we use AutoGuidance with a guidance value of 1.5, no interval, and a linear time schedule at inference.

