# OpenReview forum: "Latent Forcing: Reordering the Diffusion Trajectory for Pixel-Space Image Generation"
_ICML.cc/2026/Conference — ICML 2026 regular_

### Official Review · Reviewer_994D · 2026-03-02

**Soundness:** 3
**Presentation:** 3
**Significance:** 3
**Originality:** 3
**Overall Recommendation:** 4
**Confidence:** 3

**Summary:**

This paper introduces a multi-modular strategy that combines the latent diffusion model with the pixel diffusion model. The main claim is that we should not treat it as an either-or question. But the ordering of latent diffusion and pixel diffusion is truly important. Empirically, the paper suggests that generating latent structure earlier and refining pixels later works better. The paper also shows a bunch of experiments to support their idea.

**Compliance With Llm Reviewing Policy:**

Affirmed.

**Final Justification:**

My concerns are all addressed.

**Key Questions For Authors:**

I list below what I believe are the questions, but I would be happy to be corrected if I misunderstood any part of the work.

1. The Appendix seems not to be complete. A.1 is missing. Could the authors clarify and provide a complete appendix structure?
2. The sentence "$\alpha = 9$, with an FID-10K unguided of $18.57$" does not align with table 4, where the fid unguided for $\alpha = 9$ is $13.48$, which is correct, or am I misunderstanding?
3. In table 7, should the notation be $\mu_{\text{Latent}}$?
4. As mentioned in Section 4.2, the inference speed is stated to be nearly the same as that of DiT. Could you at least report the actual inference speed? In addition, providing runtime comparisons would give a clearer sense of how this method compares with latent diffusion in practice.

**Limitations:**

yes

**Strengths And Weaknesses:**

## strength
1. The latent diffusion model and pixel model is not an either-or question, and the design of using latent generation first and then pixel-level refinement for fine structural alignment is a natural and useful guiding strategy.
2. The paper is well written and easy to follow.
3. Compared to the DiT/JiT pipeline, the extra parameters are small, as shown in the paper.
4. The experiments are comprehensive, and many of them provide useful insights. The results also look reasonable to me.
5. I agree with the novelty of this paper.

## weakness
1. There are some typos and mismatches in the paper, see Question 1-3.
2. Parameter setup is a lot and hard to follow sometimes. For example, the epoch is sometimes 80, sometimes 200; the default $t_{\text{clip}} = 0.05$ in Sec 4.3, but changes to $1/3$ in Sec 4.5; also tricks like adding noise to the latent in Sec 4.8, etc. These inconsistencies and parameter-tuning tricks make it difficult for me to be fully confident that the improvement in results is due only to ordering. Although I believe that ordering is important.

---

> ### Author Rebuttal · Authors · 2026-03-31
>
> We thank you for your thoughtful review of our work. We are glad to see that you find our approach “natural and useful” and our paper “well written and easy to follow” while demonstrating “comprehensive” experiments.
>
> **Parameter setup.**
>
> We apologize for any confusion in the parameter setup in our approach. We believe that this difference stems from our model investigating two approaches into ordering, discussed in Sec 4.4, a “Multi-Schedule Model” and a “Single-Schedule Model.” The Multi-Schedule Model trains jointly on multiple noise schedules, allowing ordering to be determined entirely at inference. Meanwhile, the Single-Schedule model trains and tests on a single inference order. As discussed in Sec. 4.4, we include both approaches to control for different experimental settings, however this results in two sets of parameters. The Multi-Schedule allows experiments to hold model weights constant, ensuring ordering is the sole cause of performance differences, while the Single-Schedule model allows for baseline comparisons by more closely matching the training and inference distributions. We will change this split from a subsection into a full section to better delineate which parameters correspond to which approach, and we thank you for raising this concern.
>
> **Key Questions**
>
> **1. Formatting Issues**
>
> Thank you for pointing out the formatting issue in Appendix A.1. There is no missing content, however “A.1. Implementation Details” is misformatted where it should instead be a heading to subsections A.2 and A.3. We will fix this issue in our final manuscript.
>
> **2.** $\alpha = 9$  **FID-10k**
>
> The FID-10K of 18.57 corresponds to the Multi-Schedule model, Fig 3. while the FID-10K of 13.48 corresponds to the Single-Schedule model. We apologize for the confusion and will ensure that our previously addressed section split will further clarify this in an updated version of the paper.
>
> **3. Table 7 Label**
>
> Yes in Table 7 the column should be titled $\mu_\text{Latent}$, thank you for discovering the typo!
>
> **4. Inference Speed**
>
> Because diffusion integration takes up the vast majority of model forward executions for both pixel and latent diffusion, generally all diffusion approaches with the same model size, guidance, and number of diffusion timesteps will perform similarly in wall clock time. We implement your requested performance evaluation. We find that LF-DiT/L with CFG runs at **4.32 img/sec/GPU** on an H100 for generation with 50 Heun Steps, averaged over 10k samples. In the same setting, JiT/L runs at **4.29 img/sec/GPU**, meaning that Latent Forcing runs at the roughly the same speed as JiT.
>
> We thank you again for your review and hope we have managed to fully address to all questions and concerns.

---

> > ### Author Rebuttal · Reviewer_994D · 2026-04-03
> >
> > Thanks for the rebuttal. My concerns have been addressed. I maintain my positive rating.

---

### Official Review · Reviewer_1rao · 2026-03-09

**Soundness:** 3
**Presentation:** 3
**Significance:** 3
**Originality:** 3
**Overall Recommendation:** 4
**Confidence:** 3

**Summary:**

This paper proposes a latent-pixel dual stream diffusion architecture.


The proposed method orders the denoising trajectory by jointly processing latents and pixels, with separately tuned noise
schedules.

This allows the latents to act as a scratchpad for intermediate computation before high-frequency pixel features are generated.

Multi-Schedule Model trained with independently sampled time variables per modality acts like a super net in weight-sharing neural architecture search, which is used to explore the space of ordering.

**Compliance With Llm Reviewing Policy:**

Affirmed.

**Final Justification:**

My concerns are all addressed.

**Key Questions For Authors:**

(1) Will the proposed latent-pixel diffusion architecture cause extra computation overhead compared with latent only or pixel only models?

(2) How the proposed method perform on larger scale diffusion models and higher image resolutions like stable diffusion/flux 480p or 720p?

**Limitations:**

No.

Diffusing both the latent and the pixel may cause extra computational overhead.

**Strengths And Weaknesses:**

Strength:
Diffusing both the latent and the pixel is novel.

Exploring the ordering space by a supernet-like multi-schedule model trained with independently sampled time variables per modality is novel.

The manuscript is well written and easy to follow.

The improvement of FID over baseline model JIT and JIT REPA is significant.




Weakness:

Computational overhead needs to be analyzed and reported. Compared with baseline pixel or latent space models, the proposed method may cause extra computation cost because both latent and pixel are diffused with their own time shcedule.

Lacking experiments on larger scale diffusion models and image resolutions, for example, 480p or 720p with stable diffusion/flux.

---

> ### Author Rebuttal · Authors · 2026-03-31
>
> We thank you for your thoughtful review. We appreciate how you found our approach “novel” as well as our paper “well written and easy to follow” while demonstrating “significant” improvements over the baseline.
>
> **Computational overhead**
>
> Our approach does not cause significant additional computational overhead versus the baseline. Although we diffuse both latents and pixels, in our evaluations we hold the global number of timesteps constant, leading to the same number of forward executions. Specifically, we use 50 Heun steps for all experiments, leading to 100 forward passes for all models without guidance and 200 forward passes with guidance. We additionally note that at inference, the pretrained model encoder is entirely discarded and there is no decoder, meaning that diffusion integration at inference accounts for all compute performed.
>
> In practice, we find that with classifier free guidance (CFG), our Latent Forcing Large model (LF-DiT/L) obtains **4.32 images/second/GPU** on a single H100, similar to JiT/L, which obtains **4.29 images/second/GPU**, averaged over 10k samples. Therefore, our proposed approach is **roughly identical in wall-clock inference time**, if not marginally faster.
>
> **Larger scale diffusion models and higher image resolution.**
>
> We further scale our large latent forcing model to 600 epochs, increased from 200 evaluated in the paper. At 600 epochs, our approach continues to outperform JiT, where LF-DiT/L obtains a guided FID-50K of **1.93** versus JiT/L which obtains **2.36**. We are unfortunately unable to scale further to the greater H (953M) or G (2B) parameter sizes due to compute constraints.
>
> Following your request, we also train our approach at a higher image resolution, specifically the 512p setting on ImageNet. We find that LatentForcing shows benefits over JiT in this setting as well, where at 80 epochs, LF-DiT/B obtains an unguided FID-50K of **17.46** significantly outperforming JiT, which obtains an unguided FID of **51.67**.
>
> We hope that our response has addressed all of your questions and we thank you again for your review of our work.

---

> > ### Author Rebuttal · Reviewer_1rao · 2026-04-02
> >
> > My concerns are all addressed.

---

### Official Review · Reviewer_c8D5 · 2026-03-09

**Soundness:** 3
**Presentation:** 3
**Significance:** 3
**Originality:** 4
**Overall Recommendation:** 4
**Confidence:** 4

**Summary:**

This paper proposes Latent Forcing, a modification to diffusion models that aims to combine the advantages of latent diffusion and pixel-space diffusion. The key idea is to jointly process latent representations and pixels during diffusion with separately tuned noise schedules. In this design, the latent representation is revealed earlier in the denoising trajectory and acts as a temporary “scratchpad” that conditions the later generation of high-frequency pixel details. The authors argue that the order in which information is generated during diffusion is an important design dimension. By revealing latent features before pixels, the proposed approach aims to preserve the training efficiency benefits of latent diffusion while avoiding the information loss caused by a separate tokenizer and decoder.

**Compliance With Llm Reviewing Policy:**

Affirmed.

**Final Justification:**

The authors have addressed all my concerns.

**Key Questions For Authors:**

(1) The paper argues that latent representations serve as a scratchpad that facilitates pixel-space diffusion. However, it is unclear why such a draft is necessary. Why can pixel-space diffusion not directly generate coarse image structures from noise during early denoising steps? A clearer explanation of the theoretical motivation for introducing the latent representation would be helpful.

(2) The proposed method relies on a pretrained latent encoder. To what extent do the improvements come from the proposed latent forcing mechanism rather than the pretrained encoder itself? An ablation isolating the effect of the encoder would help clarify this point.

(3) The paper suggests that latent representations should be generated before pixel details. Would it be possible to instead perform diffusion in latent space during early timesteps and then switch to pixel-space diffusion for later steps without joint training? How would such a two-stage strategy compare with the proposed method?

(4) The reported improvement over the JIT baseline (FID 2.63 vs 2.79) is relatively small. Additionally, the reported JIT-L FID appears inconsistent with the value (2.36) in the original paper, and could the authors provide more comparisons with JIT across different model scales (e.g., B, H, and G)?

**Limitations:**

No. I recommend that the authors include a section outlining potential limitations, such as the reliance on pretrained latent encoders, the scalability to larger datasets, and any failure modes observed. Additionally, a brief discussion of the broader impact of generative models in terms of misuse, bias, or ethical concerns would strengthen the manuscript.

**Strengths And Weaknesses:**

Strengths

(1) The authors propose a method that jointly models latent space and pixel space to perform pixel diffusion, and systematically investigate how the generation order between these two representations affects performance. This perspective provides a new viewpoint for understanding and improving pixel-space diffusion models.

(2) The paper presents relatively comprehensive experiments, and the conclusions drawn in the work are generally supported by empirical evidence.

(3) The proposed method is relatively simple and practical. It can be easily integrated into existing diffusion frameworks and demonstrates measurable performance improvements.

Weaknesses

(1) Although the authors argue that the latent representation can serve as a scratchpad which is subsequently refined by pixel-space diffusion to generate high-frequency details, the paper does not sufficiently explain the theoretical motivation behind this design. In particular, it remains unclear why pixel-space diffusion necessarily requires such a draft in order to operate effectively, and why the model cannot directly generate coarse image structures from pure noise during the early stages of diffusion. The necessity of introducing the latent representation therefore requires further clarification.

(2) Compared with pure pixel-space diffusion methods, the proposed approach relies on a pretrained latent encoder. This requirement partially reduces the practical simplicity of the approach. Moreover, the observed performance improvement may largely stem from the pretrained knowledge contained in the encoder, which somewhat contradicts the original motivation of pixel-space diffusion methods that aim to avoid additional representation learning modules.

(3) The authors conclude that the generation process should first produce latent representations and then generate pixel-space details. This raises a natural question: would it be possible to first perform diffusion in the latent space during the early portion of the trajectory (e.g., t \in [0, 0.5]), and then switch to pixel-space diffusion to generate the final image? Such a two-stage approach could similarly provide a coarse draft through latent representations while allowing pixel diffusion to refine details. It would be interesting to compare this simpler strategy, which does not require joint training, with the proposed method to understand whether comparable improvements can be achieved while maintaining the advantages of pixel-space diffusion.

(4) The final experimental results show an improvement from the baseline JIT method (FID = 2.79) to FID = 2.63, which is relatively modest. In addition, the reported FID for JIT-L appears to be inconsistent with the value reported in the original paper, and further clarification regarding the experimental setup would be helpful.

---

> ### Author Rebuttal · Authors · 2026-03-31
>
> We thank you for your thoughtful review. We appreciate how you believe our approach “provides a new viewpoint for understanding and improving pixel-space diffusion models” while reporting “comprehensive experiments.” We are glad you agree our approach is “simple and practical” while attaining “measurable performance improvements.”
>
> We group corresponding Weaknesses and Key Questions (KQ) in our rebuttal.
>
> **Weakness/KQ 1: Theoretical Justification:**
>
> We agree that theoretical motivation is important, and there is a deep relation of ordering to learnability. Discussed in Sec. 2, [152-155] “one especially important theoretical property of modifying information order in diffusion generative models is that different conditioning orderings can lead to exponential differences in the learnability of data (Gupta et al., 2024).” The argument of Gupta et al. follows from the irreversibility of one-way functions, where diffusing the input before the output of a one-way function is tractable but the reverse is not. This reveals that different orderings of diffusion, explored in our paper as generating features before or after pixels, can have provable effects on learnability. We dedicate the rest of our work to demonstrating that this ordering significantly improves learnability in practice. We will expand on this theory in our updated paper.
>
> **Weakness/KQ 2: Are the improvements from ordering rather than the encoder?**
>
> **Yes**, in our work we demonstrate with several experiments that gains are purely from order. Most directly, we address this in Sec. 4.11: “Distillation vs Ordering” where we contrast REPA feature distillation with the feature ordering in Latent Forcing: “we demonstrate that the gains from ordering are distinct from distillation, with a 1.9× reduction in unguided FID-50K versus JiT with REPA, and a 2.5× reduction compared to JiT.”
>
> We show complimentary results in Fig. 3 and Tab. 1, where solely modifying the inference time trajectory of our Multi-Schedule model, holding constant model weights and the use of a pretrained encoder, leads to large improvements: from 56.57 to 18.57 FID-10K, Fig. 3. Finally, we show in Table 1 that ordering applies to several latent spaces, including Data2Vec2 and a 64x64 downsampled image which uses *no pretrained model*.
>
> **Weakness/KQ 3: “Would it be possible to instead perform diffusion in latent space during early timesteps and then switch to pixel-space diffusion for later steps without joint training?”**
>
> **Yes**, and we implement this in our work as the “Cascaded” model (Sec 4.7, 4.8), finding it achieves the best performance (Table 4). In Latent-Forcing, we train two model types, discussed in Sec 4.4. The “Multi-Schedule Model” jointly trains on timestep combinations for features and pixels, while the “Single-Schedule Model” trains and evaluates using a single trajectory without jointly training. In the Cascaded Single-Schedule Model, the first 50% of timesteps diffuse latents, and the last 50% diffuse pixels. We also find that adding noise at training to the latents during pixel diffusion timesteps further improves performance versus the approach you mentioned (Tables 2,3).
>
> **Weakness/KQ 4: FID evaluation**
>
> **“Inconsistent” FID:** We obtain the 2.79 FID value from Table 6 in JiT (arxiv.org/pdf/2511.13720), which reports JiT/L at 200 epochs. 2.39 refers to JiT/L at 600 epochs. In our work, we report ablations at 80 epochs and compare baselines at 200 epochs due to compute constraints.
>
> **Additional sizes:** Following your request, we train a new base-scale Latent Forcing model (LF-DiT/B). We also train LF-DiT/L to 600ep. Due to compute constraints, we are unable to scale to H (953M) or G (2B). At these new scales, our model continues to outperform JiT:
> * LF-DiT/B obtains 15.26/6.77 FID-50K outperforming JiT/B with 46.28/8.92 at 80ep unguided/guided respectfully
> * LF-DiT/L obtains 4.74/1.93 FID-50K outperforming JiT/L 13.70/2.36 at 600ep unguided/guided respectfully
>
> **Marginal Improvement**: We report many comparisons against JiT *all with improved performance*. In Tab. 11, JiT/L obtains an unguided FID-50K of 16.21 versus 7.2 with LF-DiT/L, a **2.2x** improvement, as well as the referenced 2.79 to 2.63 decrease with guidance. In Table 10, for unconditional generation, guided/unguided FID-50K decreases from 53.26/44.80 with JiT/L to 20.44/13.16 with LF-DiT/L. Finally, at the newly evaluated 600ep, LF-DiT/L obtains a guided FID-50K of 1.93 vs 2.36 from JiT/L.
>
> **Limitations**
>
> Although our method applies to several encoders, an encoder must still be trained, which may not be readily available in other domains. Further, while Latent Forcing is empirically demonstrated on ImageNet, we hope that future work expands it to additional datasets. Our work operates in the area of image generative models, which like all generative models, can lead to harmful dual-use behavior, such as Deepfakes. We will include and expand upon all of these points in our updated paper.

---

> > ### Author Rebuttal · Reviewer_c8D5 · 2026-04-04
> >
> > Thank you for the authors’ response. My confusion regarding the FID metric has been resolved, but I still have the following concerns:
> >
> > The paper does not provide a rigorous analysis of the learnability of pixel-space diffusion models under different noise levels. In particular, it is not clearly explained why pixel diffusion becomes difficult at high noise levels and why assistance from latent diffusion is necessary.
> > Replacing latent features with 64×64 downsampled images leads to a significant drop in performance, which further suggests that the proposed method relies heavily on pretrained encoders.
> > The authors may have misunderstood my question. I am asking whether the following training-free approach could also improve pixel-space diffusion without introducing reconstruction loss: first generate an image using a pretrained latent diffusion model, then decode it into pixel space, add noise again, and finally apply a pixel-space diffusion model for denoising.

---

> > > ### Author Response · Authors · 2026-04-05
> > >
> > > **Reliance on Pretrained Encoders**
> > >
> > > There are two separate issues here that we suggest disentangling:
> > >
> > > Issue 1: Does latent forcing require a pretrained encoder?
> > >
> > > The answer here is a clear **no** – We demonstrated that the main technique of latent forcing—reordering the diffusion trajectory—leads to improved generation performance, even with *no pretrained encoder*. In Table 1 of the main paper, we treat downsampled 64x64 images as the latent space and show that generating in an ordered trajectory yields a 2.2 FID improvement over generating full-resolution pixels first. While pre-trained encoders can get further benefits, this experiment shows that diffusion trajectory reordering is impactful, even in isolation.
> > >
> > > Issue 2: Does relying on a pre-trained encoder diminish the significance of latent forcing?
> > >
> > > We have argued above that latent forcing shows benefits even in the absence of a pretrained encoder, suggesting that ordering itself can lead to benefits even without any external models.
> > >
> > > However, we argue further that **even if** latent forcing did only show benefits when paired with an external encoder, it would still be of significant interest to the community and worthy of publication. Using an external encoder should not be seen as a disqualifying factor.
> > >
> > > The authors agree with the reviewer that we should prefer methods that do not require external encoders. However, methods such as REPA, RAE, etc which do require them have had enormous impact on the community, and have become nearly ubiquitous in SOTA recipes. The community has thus demonstrated incontrovertibly clear interest in different methods for generative modeling that improve performance by relying on external encoders in some way. Latent forcing (when viewed in its most uncharitable form) is a novel method by which generative models can benefit from external encoders, with clear benefits over REPA. As such, we believe that it would be of broad interest to the community.
> > >
> > > **Cascaded Baseline**
> > >
> > > We have run the additional baseline suggested by the reviewer:
> > >
> > > **Experiment Setup:** We use a ViT/L latent diffusion model on DINOv2 with a ViT/L diffusion decoder that converts DINOv2 to pixels, and a JiT/L as the pixel space decoder. We evaluate on FID-10K with and without guidance at 80 epochs. Importantly, this baseline results in checkpoints that train with 3x the GPU compute (80 epochs for latent diffusion, 80 epochs for latent-to-pixel decoding, and 80 epochs for JiT/L pixel diffusion) as Latent Forcing. We stratify our evaluation into different noise levels, 25%, 50%, and 75%, at which the transition from latent to pixel diffusion occurs. For example in the 25% setting, we 1) fully diffuse latents, 2) decode latents to pixels, 3) noise to t=0.25 (t=0 is noise), and 4) continue with pixel diffusion for the remaining 75% of timesteps. For all other implementation decisions, we follow the same setup as Latent Forcing (e.g. 50 Heun steps split evenly between latent/pixel diffusion).
> > >
> > > | Model / Configuration | JiT/L | Baseline 25% | Baseline 50% | Baseline 75% | LF-DiT/L |
> > > | :--- | :---: | :---: | :---: | :---: | :---: |
> > > | **Unguided FID-10K** | 28.26 | 25.15 | 22.00 | 19.13 | **12.42** |
> > > | **Guided FID-10K** | 9.67 | 8.12 | 8.27 | 8.22 | **6.36** |
> > > | **Training Epochs** | **80** | 80+80+80 | 80+80+80 | 80+80+80 | **80** |
> > >
> > > We observe that **Latent Forcing obtains the best performance** both with and without guidance, while using 3x less compute than the proposed baseline during training.
> > >
> > > **Explanation of Why Latents Beat Pixels**
> > >
> > > We dedicate Figures 4 and 5 in our work to analyzing why latents beat pixels at high noise levels, demonstrating that noisy latents encode greater structural detail than noisy pixels at equivalent PSNR. In Sec 4.5 [299-304], we qualitatively evaluate observations in single-step generations of DINO and Pixel features: "Even though both [DINOv2 and pixel conditioned] reconstructions have the same MSE on the ground truth image, the DINOv2 predictions maintain significantly more structural features. Meanwhile, the pixel generation component demonstrates large-scale structural uncertainty." That latents emphasize structure, and structure improves generation performance, is corroborated by recent work [1] finding that structural information serves as the best target for distillation.
> > >
> > > [1] https://arxiv.org/abs/2512.10794

---

### Official Review · Reviewer_zKJr · 2026-03-12

**Soundness:** 4
**Presentation:** 4
**Significance:** 3
**Originality:** 3
**Overall Recommendation:** 5
**Confidence:** 4

**Summary:**

The paper presents Latent Forcing, a novel training and inference strategy for pixel-space diffusion models that combines the efficiency of latent diffusion with the end-to-end benefits of modeling raw pixels.
The key idea is to train a single DiT model that can process both noised latent and noised pixel, each with its corresponding noise schedule.
Latent Forcing schedules the denoising process to reveal global structure in the latents before high-frequency pixel details, the latents act as a sort of a "thinking" mechanism found in the NLP literature, that conditions and simplifies the pixel-space generation.
The authors demonstrate that the order of conditioning is a driver of performance, ultimately achieving state-of-the-art results for pixel-space DiTs on ImageNet-256.

**Compliance With Llm Reviewing Policy:**

Affirmed.

**Final Justification:**

The rebuttal has resolved my minor comments. My assessment of the paper remains positive, and I am keeping my original score of 5.

**Key Questions For Authors:**

* Since its seems that the cascade schedule works best (Tab.1), did the authors consider the connection of their approach to the "thinking" mechanism in NLP?


* As the latents serves as a "scratchpad" for the pixels generation, did the authors found that their proposed approach helps improve condition alignment as well as generation quality?

**Limitations:**

yes

**Strengths And Weaknesses:**

Strengths:

* The writing is clear which make the paper easy to follow.
* The paper presents a novel idea while keeping the implementation conceptually simple.
* The authors provide many experiments and analysis of their proposed approach.


Weaknesses:

* No major weaknesses in the paper. See questions below.

---

> ### Author Rebuttal · Authors · 2026-03-31
>
> We thank you for your review and are glad to see that you appreciate our work. In particular, we are happy to see that you find our paper “easy to follow” and “novel” while being “conceptually simple” and with “many experiments and analysis.” We are happy to hear that you found “no major weaknesses” in our paper.
>
> **“Did the authors consider the connection of their approach to the "thinking" mechanism in NLP?”**
>
> Yes, we believe that there is a strong connection between Latent Forcing and chain-of-thought generation in NLP. In particular, in the Serial Scaling Hypothesis [1], it’s proven that increasing diffusion timesteps, although seemingly a path to inference time computation, provably cannot lead to test time scaling in the same way that chain of thought can in NLP. Specifically, multistep reasoning is limited by the capacity of the network rather than chain of thought which can obtain unbounded length reasoning by increasing token count. We view Latent Forcing as a preliminary attempt at introducing image-native chain-of-thought like test-time-scaling to diffusion models. In particular, we see Latent Forcing as introducing a fixed-size “scratchpad”, akin to allowing a language token a finite number of additional number of reasoning tokens before outputting an answer. We hope that the extremely important question of how to introduce unbounded general-length chain-of-thought like compute into diffusion can be addressed in future work.
>
> **Condition alignment**
>
> Yes, using Inception Score IS as a proxy for condition alignment and FID as a proxy for generation quality, we find that introducing a latent scratchpad improves IS, particularly in unguided settings where class alignment is not assisted by guidance. For Latent Forcing at Large scale (LF-DiT/L), we obtain an IS of 167.5/306.8 without and with guidance respectfully, outperforming JiT/L with 104.6/298.5. We consistently find that the gains from Latent Forcing can lessen the need for class guidance for prompt alignment in our paper.
>
> We thank you again for your review.
>
> [1] https://openreview.net/forum?id=ObXB7KJn0B

---

> > ### Author Rebuttal · Reviewer_zKJr · 2026-04-03
> >
> > I appreciate the rebuttal by the authors. I decide to maintain my original score.

---

### Decision · Program_Chairs · 2026-04-30

**Decision:**

Accept (regular)

**Comment:**

The paper initially received 1 Accept, 2 Weak Accept, and 1 Weak Reject. The rebuttal addressed the reviewers' concerns well. Hence, Reviewer c8D5 increased his score from Weak Reject to Weak Accept, making all final scores favor paper acceptance. The reviewers praised the work for its clear writing, novel idea, and comprehensive experiments with strong results.

The ACs reviewed and agreed with the acceptance decision.